# Response of the Fate of In-Season Fertilizer Nitrogen to Plastic Mulching in Rainfed Maize Croplands of the Loess Plateau

**DOI:** 10.3390/plants11182343

**Published:** 2022-09-08

**Authors:** Xueli Zhang, Bin Hu, Shangwen Wang, Wenyi Dong, Subramaniam Gopalakrishnan, Tao Jin, Enke Liu

**Affiliations:** 1Institute of Environment and Sustainable Development in Agriculture, Chinese Academy of Agricultural Sciences, Beijing 100081, China; 2Shandong Agricultural Technology Promotion Center, Jinan 250100, China; 3International Crops Research Institute for the Semi-Arid Tropics (ICRISAT), Patancheru 502324, India; 4State Key Laboratory of Hulless Barley and Yak Germplasm Resources and Genetic Improvement, Lhasa 850002, China

**Keywords:** plastic mulch, fertilizer nitrogen recovery, soil nitrogen, nitrogen concentration in organ

## Abstract

As plastic mulching is widely used for maize production on Loess Plateau, study of the fate of fertilizer nitrogen (N) in rain-fed croplands is of great significance. Field experiments were conducted during 2015–2016 at a typical dry-land farm on the Loess Plateau, China. The stable isotope tracer technique was applied to analyze the effects of plastic mulching on the maize crop yield, N content in the grain, and mechanism of N uptake and utilization in maize plants with plastic mulch (PM) and without plastic mulch (CK) on the Loess Plateau. Maize yield, aboveground dry matter, grain N concentration, and N uptake in aboveground biomass for PM significantly increased, in addition to fertilizer nitrogen recovery and nitrogen production efficiency. Compared to CK, PM improved the total N uptake from the soil in the aboveground biomass by 16.39 and 27.75 kg ha^−1^ and fertilizer nitrogen recovery by 10.89 and 22.02 kg ha^−1^, respectively. Furthermore, PM increased in-season fertilizer N retention in the soil by 11.9–24.8 kg ha^−1^, and the uncountable fertilizer N decreased by approximately 33.8 kg ha^−1^ on average. In conclusion, PM simultaneously improved the maize yield and N utilization, which provides a scientific basis for nitrogen management in maize croplands.

## 1. Introduction

Dry-land agriculture accounts for more than 70% of the total farmland in northern and northwestern China, which includes the vast rain-fed areas to the north of the Qiling Mountains and the Huaihe River [1], and plays an essential role for food security in China. Water shortages and nutrient deficiencies are the two major restrictive factors for food production in dry-land farm systems [2,3]. Plastic mulch has beneficial effects on crop production, such as preserving soil heat and water, and results in an increase in water use efficiency, particularly in dry-land agriculture [4,5].

Maize (*Zea mays* L.) is one of the major food crops on the Loess Plateau, accounting for 27.3% of the total agricultural area [6], and it is the first food crop for which plastic mulching was used. The extended area of maize with plastic mulching is over 0.2 million ha on the Loess Plateau [7].

Maize is sensitive to nitrogen availability, and crop yield has a positive correlation with in-season fertilizer N uptake [8]. In maize production, the yield and quality of the crop depend to a large extent on nitrogen (N) management. Unsound N management is more likely to cause environmental pollution, including water contamination and atmospheric pollution. Plastic mulching increases maize yield and simultaneously changes maize nitrogen uptake. A better understanding of the fate of in-season fertilizer under plastic mulch is crucial to an extensive dry-land farm.

Many previous studies have analyzed the effects of plastic mulching on soil temperature, seed germination, root growth, crop yield, and water use efficiency [9,10]. However, the effects of plastic mulch on the fate of fertilizer N and various parameters related to the efficient use of nitrogen still need to be further studied. Wang et al. [11] found that 67.6–73.2% of the nitrogen in maize plants was derived from the soil under plastic mulching, and Liu et al. [12] reported that the maize nitrogen uptake from the soil increased under plastic mulching, but plastic mulching decreased in-season labeled-N uptake by 19%.

In our analysis of the effect of plastic mulching on maize nitrogen use mechanisms, we hypothesized that (1) the plastic film would increase the crop nitrogen uptake and fertilizer N retention in the soil, and (2) plastic mulch would increase the transfer of nitrogen from nutritive organs to maize kernel. To test these hypotheses, we conducted a field experiment in a rainfed maize cropland on the Loess Plateau of China. A ^15^N stable isotope tracer technique was employed to analyze the response of the characteristics of maize nitrogen uptake, the distribution of fertilizer N in the maize organs, and the fate of in-season fertilizer N under plastic mulching to elucidate the mechanism of maize nitrogen uptake and utilization, and the fate of in-season fertilizer N, which could provide a scientific basis for the reduction of fertilizer N loss, the improvement of the fertilizer nitrogen recovery, and sustainable rain-fed agriculture.

## 2. Results

### 2.1. Yield and Aboveground Dry Matter

Maize grain yield and aboveground dry matter were both significantly affected by film mulching at harvest in 2015 and 2016, respectively (Figure 1). The leaves, stem, and husk leave dry matter of the PM treatment were significantly higher than that of the CK treatment by 22.8%, 34.2%, and 12.3% in 2015, respectively, and the leaves, stem, cob core, and husk leave dry matter in the PM treatment increased by 17.6%, 22.7%, 28.6%, and 24.7% in 2016, respectively.

### 2.2. Accumulation and Distribution of Nitrogen in Spring Maize

#### 2.2.1. Nitrogen Concentration of Aboveground Parts in Maize at HARVEST Time

There was a significant difference (*p* < 0.05) between CK and PM for the total nitrogen concentration of different aboveground organs (Figure 2). Compared to CK, PM increased the nitrogen concentration in the grain by 12.09% and 7.16% for both years, respectively. The order (from high to low) of the nitrogen concentration in the different organs was grain (leaves) > cob cores > husk leaves > stems. There were no significant differences in the nitrogen concentration of the leaves, husk leaves, and stems for both treatments (*p* > 0.05).

#### 2.2.2. Total N Uptake in the Grain and Aboveground Biomass

Compared with CK, PM significantly increased the total nitrogen uptake in the grain and the aboveground biomass by 19.3 and 27.3 kg ha^−1^ in 2015 and 48.0 and 49.9 kg ha^−1^ in 2016, respectively (Figure 3).

#### 2.2.3. Sources of Nitrogen in Aboveground Organs

The nitrogen in the aboveground biomass was mainly distributed in the grain. When mulch was used compared with no mulch, the total N uptake from the soil and fertilizer in the grain was significantly increased by 9.29 and 10.03 kg ha^−1^ in 2015 and 32.16 and 15.88 kg ha^−1^ in 2016, respectively (Figure 4). Compared to CK, PM increased the nitrogen uptake in the grain from the soil and fertilizer by 9.2% and 27.9% in 2015 and 32.8% and 37.3% in 2016, respectively.

Cultivation with film-mulching (PM) increased the total N uptake from the soil in the aboveground biomass by 16.39 and 27.75 kg ha^−1^ in 2015 and 2016, respectively, and the in-season fertilizer N uptake by 10.89 and 22.02 kg ha^−1^ in 2015 and 2016, respectively, compared to CK (Figure 5). Equivalently, PM increased the plant soil N uptake ratio by 8.4% and 15.5% and the plant fertilizer N uptake ratio by 19.9% and 32.2%, respectively, for both years.

### 2.3. Nitrogen Efficiency

PM increased the fertilizer nitrogen recovery, the nitrogen uptake efficiency (NU_p_E), and the nitrogen production efficiency (NPE) (Table 1). There was no significant difference between PM and CK in NHI in 2015 and 2016, respectively, but NHI in 2016 was larger than in 2015 under the identical treatment. Compared to CK, PM increased NPE by 12.8% and 25.3% for 2 years. PM had no significant effect on NU_p_E in 2015 but increased it by 29.5% in 2016. PM had no significant effect on the nitrogen harvest index but increased fertilizer nitrogen recovery by 19.9% and 32.2%, respectively, in 2015 and 2016.

### 2.4. The Fate of Fertilizer N

The fertilizer nitrogen budget based on the fate of ^15^N-urea in the upper 60 cm of the soil and labeled-N at maturity is shown in Table 2. PM significantly increased the in-season maize uptake and residual N in the soil. Nitrogen uptake from in-season fertilizer nitrogen in the aboveground biomass accounted for 29.1% and 40.2% of the applied fertilizer for PM in 2015 and 2016, respectively. Compared to CK, PM improved the total N uptake from the soil in the aboveground biomass by 16.39 and 27.75 kg ha^−1^. Compared to CK, PM increased the residual N in the soil (0–60 cm) by 29.5% and 15.0% in 2015 and 2016, respectively. PM significantly decreased the unaccounted-for fertilizer N, which mainly included the nitrogen fertilizer 60 cm below the soil, the nitrogen fertilizer lost in the farmland, and nitrogen gas. The unaccounted-for fertilizer N comprised 33.9–37.6% of the applied fertilizer under the CK treatment and 18.9–22.6% under PM treatment in 2015 and 2016, respectively Compared to CK, PM improved fertilizer nitrogen recovery by 10.89 and 22.02 kg ha^−1^, respectively. Furthermore, PM increased in-season fertilizer N retention in the soil by 11.9–24.8 kg ha^−1^, and the uncountable fertilizer N decreased by approximately 33.8 kg ha^−1^ on average.

## 3. Discussion

### 3.1. Yield and Aboveground Dry Matter

Our data showed that PM increased the maize grain yield by 12.8% and 25.3% in 2015 and 2016, respectively. Our results are consistent with findings in the current literature [13,14]. Li et al. [15] found that plastic film increased the maize yield by 8–25% on the Loess Plateau. Liu et al. [12] found that plastic mulching improved the maize yield by 70–72% in semi-arid northwest China. The effects of plastic mulching on crop yield show obvious regional differences [13]. In some areas with limited water and appropriate temperature, plastic mulching could significantly increase crop yield [16,17]. In semiarid Kenya, maize productivity has been unsatisfactory due to insufficient use of rainwater, and plastic mulching increased maize yield by 66.5–349.9% [18]. In the rain-fed croplands of northern China, water shortages and low temperature are restrictive factors that affect crop productivity [16,19]. Plastic mulching improved the root system and increased the soil water content and nutrient uptake by increasing the soil water and temperature [10,20], facilitating the transport of more assimilated N into the grain, and closing the yield gap between the attainable and actual yield, ultimately resulting in a higher grain yield [21]. Plastic mulching improved the N uptake and nitrogen use efficiency by increasing the accumulation ability of corn kernels and dry matter.

### 3.2. The Absorption and Utilization of Nitrogen

Our study found that plastic mulching increased the nitrogen concentration in corn kernels. The nitrogen concentration in the maize grain was mainly from nitrogen uptake after the maize silking stage and the reuse of accumulated nitrogen in the maize organs before the silking stage [22,23]. Xu et al. [24] found that approximately half of the total nitrogen accumulation in maize grains was from root uptake after flowering, and another half was from the other organs’ nitrogen accumulation. Plastic mulching increased the activity of enzymes associated with nitrogen transformation in the soil, thus improving the effectiveness of the soil nitrogen [25,26], and ultimately promoted and maintained nitrogen absorption and utilization before and after the maize silking stage, providing a substantial basis for nitrogen transfer to the grain. Moreover, plastic mulching benefitted the transfer of nitrogen from the nutritive organs to the grain [27] by increasing the nitrogen content in maize grains by enhancing the absorption of nitrogen by the maize roots and increasing nitrogen transport to the grain.

Compared to CK, PM significantly increased the aboveground biomass nitrogen uptake by 10.9–29.5%. Liu et al. [12,28] also found that PM boosted the aboveground biomass nitrogen uptake and increased the amount of total nitrogen accumulated by maize plants by 33.2–55.8%. Wang et al. [29] found that plastic mulching increased the aboveground biomass nitrogen uptake by 16–34% in semi-arid rain-fed maize croplands in a 3-year field experiment. However, for years when water shortage occurred in the early growth stage, plastic mulching had no significant effect on the aboveground biomass nitrogen uptake. The increase in aboveground biomass nitrogen uptake by maize due to plastic mulching was mainly associated with an increase in the soil temperature and the effectiveness of the soil nitrogen, and a decrease in soil evaporation due to plastic mulching [30,31].

### 3.3. N Efficiencies under PM

However, Liu et al. [12] found that film mulching decreased the in-season labeled-n uptake by 19%, but we obtained an opposite result. The fertilizer nitrogen recovery of PM was 19.9% and 32.2% higher than that of CK in 2015 and 2016, respectively, which agreed with the results of Deng et al. [1]. Wang et al. [8] found that the fertilizer nitrogen recovery was positively correlated with yield, so when PM increased the maize yield, it also concomitantly increased the fertilizer nitrogen recovery. More than 50% of the accumulated nitrogen from the aboveground biomass was distributed in the grain, and the NU_p_E and PM improved the fertilizer nitrogen recovery due to the increase in grain yield and the nitrogen content of the grain. In the present study, PM had no significant effects on NU_p_E in 2015; however, NU_p_E was increased by 29.5% in 2016.

### 3.4. Effects of PM on Maize N Assimilation

Does maize uptake nitrogen mainly from the soil nitrogen or fertilizer nitrogen? It is important to assess soil fertility improvement in rain-fed cultivated land under plastic mulching. Zhang [7] found that film mulching had no significant influence on in-season fertilizer nitrogen uptake compared to no mulching. Liu et al. [12] found that plastic mulching increased the maize nitrogen uptake from the soil; however, it decreased the in-season fertilizer nitrogen uptake. Our study revealed that the soil nitrogen uptake in maize plants under PM was 16.39 and 27.75 kg ha^−1^ higher than that under CK for both years, and the fertilizer nitrogen was 10.89 and 22.02 kg ha^−1^ higher under PM for the 2 consecutive years, respectively, indicating that PM improved the nitrogen uptake not only from the soil but also from the in-season fertilizer. This might be attributed to the acceleration of the mineralization of soil organic matter due to plastic mulching [31], increasing the soil inorganic nitrogen content in the plow layer, which increased the root nitrogen uptake. The increased nitrogen uptake could boost root growth and enlarge the contact area between the available nitrogen (soil inorganic N and fertilizer N) and the root system, which increased the absorption of the available nitrogen by the roots in return.

To improve the nitrogen uptake from the fertilizer and the soil, PM increased the proportion of in-season fertilizer nitrogen uptake (19.9–32.2%) significantly more than that of the nitrogen uptake from the soil (8.45–15.5%). Nitrogen sequestered by the maize plants accounted for 62–80% of the nitrogen mineralized by the soil organic matter and approximately 70% of the nitrogen in the maize plants was from the soil [11,12]. Thus, the proportion of fertilizer nitrogen uptake was significantly higher than that of the soil nitrogen uptake when the amount of nitrogen taken up was slightly increased.

### 3.5. The Fate of In-Season Fertilizer Nitrogen

The amount of fertilizer nitrogen lost in the soil is 45 kg N ha^−1^ per year, and plastic mulching is an effective way to reduce the volatilization and leaching of fertilizer nitrogen [32], enhance root nutrient uptake, and boost the fertilizer nitrogen recovery [33]. The two-year experimental data indicated both in-season fertilizer nitrogen uptake by the aboveground maize biomass and the residual amount of fertilizer nitrogen in the 0–60-cm soil layer was increased with plastic mulching in the Loess Plateau of China, regardless of whether the annual rainfall was normal or dry. The discussion on the effect of plastic mulch on fertilizer nitrogen uptake has been referenced in the previous section, so we will focus on the retention of fertilizer N in the soil. Liu et al. [12] found that the residual amount of in-season fertilizer nitrogen in the soil with plastic mulching was significantly higher than without plastic mulching due to less ammonia volatilization under plastic mulching compared to no mulching. Wang et al. [32] found that residual fertilizer nitrogen in the 0–200-cm soil layer accounted for 48.3–51.3% of the applied fertilizer nitrogen, and approximately 50% of the residual nitrogen fertilizer was in the 0–20-cm soil layer. Our data showed that more than 55% of the residual nitrogen fertilizer existed in the 0–20-cm soil layer under the PM treatment, but under the CK treatment, only 45% of residual nitrogen fertilizer existed in the 0–20-cm soil layer. Most of the labeled-N retention was in the plow layer, which might have been associated with the assimilation of fertilizer N by microorganisms and roots in the cultivated layer due to the increase in the soil temperature and soil moisture with plastic mulching. The change in the soil micro-environment accelerated microbial activity; therefore, more in-season fertilizer nitrogen was sequestrated by mineralizing microbes. Ultimately, fertilizer nitrogen under film mulching was retained in the main form of organic N, in contrast to that under CK [12]. Outside of the growing season, the interaction between labeled-N immobilization and mineralization needs to be studied.

Unaccountable fertilizer nitrogen in the plastic mulching treatment was significantly lower than in the absence of mulching, indicating that plastic mulching significantly improved the effectiveness of in-season fertilizer nitrogen. Plastic mulching is a type of physical barrier that reduces ammonia emissions [11,12]. Liu et al. [33] found that NO_3_-N under plastic mulching was 48–86% lower than without mulching in the 100–200-cm soil profile even in years with high rainfall, demonstrating that plastic mulching could decrease the loss of nitrogen from leaching. Thus, plastic mulching reduced the unaccounted-for fertilizer N and increased the fertilizer N recovery by reducing ammonia volatilization and the loss of NO_3_-N due to leaching.

## 4. Materials and Methods

### 4.1. Study Area

The field experiment was conducted in two growing seasons in 2015 and 2016 at the Shouyang Experimental Station of Dryland Agriculture and Environment, China (37°45′58″ N, 113°12′9″ E and 1202 m a.s.l.). The climate at the experimental site is a continental temperate climate with a mean daily temperature, an annual accumulated temperature, and annual frost-free days of 7.4 °C, 3200 °C (≥10 °C), and 140 d, respectively. The inter-annual rainfall showed great variation, with a mean annual rainfall of 481 mm, and the monthly rainfall distribution was uneven and mainly distributed from June to September, which accounted for 70% of the annual rainfall. The annual mean rainfall during the growing season (1 May to 30 September) was 427 mm during the last 30 years; the precipitation during the growing season was 337 mm in 2015 and 401 mm in 2016.

The soil at the experimental site is loam according to the USDA texture classification system, and it is classified as a calcaric cambisol according to the world reference base for soil resources [34]. Before fertilization in 2015, the soil contained 0.85 g total N kg^−1^, 18.21 g organic matter kg^−1^, 84.30 mg available nitrogen kg^−1^, and 7.4 mg Olsen P kg^−1^ in the top 20 cm soil, and the soil pH was 8.1 on average (water: soil, 2.5:1). The soil bulk density ranged from 1.20 to 1.26 g cm^−3^.

### 4.2. Experimental Design

In the main experiment, two treatments were replicated three times in a randomized complete block design. The two treatments were (1) flat-planted maize without mulching (CK) and (2) flat-planted maize with partial plastic mulching (75%) (PM), and there was a gap of 25 cm between 2 film strips. The plastic film used in the experiment was colorless and transparent. Each strip was 0.008 mm thick and 75 cm wide. Each of the 6 plots (hereafter referred to as the “main plots”) was 48 m^2^ (6 m × 8 m). Each main plot was fertilized with urea granules at a rate of 225 kg N ha^−1^, superphosphate 70 kg P_2_O_5_ ha^−1^, and potassium sulfate 50 kg K ha^−1^, and only once fertilization was performed before plowing.

Micro plots were established to monitor the fate of ^15^N from the applied fertilizer N in the present study by a ^15^N isotope tracing technique. For the ^15^N labeling study, a rectangular micro plot (120 cm × 60 cm; 0.72 m^2^) was established inside of each main plot 2 m from the sides of the main plot before the main plots were fertilized in 2015, and the new micro plots were established in the same main plots in 2016. Each micro plot was tightly enclosed by 4 glass plates (7 mm thick) to a depth of 60 cm. The soil inside the micro plots was discretely protected from contamination by N fertilizer from the other main plots. Within each micro plot, 34.69 g of ^15^N-labeled urea granules (containing 46.7% N with 10.19% ^15^N abundance) equivalent to 225 kg N ha^−1^ (identical to the main plot application rate) was spread on the soil surface before plowing. After emergence, 4 maize plants were kept in each micro plot in each treatment to match the maize density of the main plots. All the other management practices for the micro plots were the same as for the main plots.

The local cropping system is one crop per year, and the main crop is spring maize. The experimental site is a rain-fed maize cropland, for which water consumption totally depends on precipitation, soil water holding capacity, and MPA. A maize crop designated Jingdan-951 was sown on 1 May and harvested on 27 September in 2015 and sown on 4 May and harvested on 28 September in 2016. The row spacing was 50 cm, and the plant spacing was 30 cm for a total plant density/Ha of 66,667 under local current commercial crop production conditions.

During both cropping seasons, weeds that grew in the non-mulched micro and main plots were pulled out manually at the early seedling stage and then returned to the cropland. In the mulched micro and main plots, the weeds under the plastic film were left unmanaged, but if the weeds grew outside the film, the worker managed them as for the non-mulched plots. No chemicals were used to control diseases or pests.

### 4.3. Sampling and Measurements

Soil sampling: Before sowing and after harvest, the soil in each micro plot and main plot was sampled using an auger (40 mm inner diameter) at depths of 0–10, 10–20, 20–30, 30–40, 40–50, 50–60, 60–70, 70–80, 80–90, and 90–100 cm. The soil was sampled from three points in each plot. Each soil sample was divided into three sub-samples, except for the roots and the small gravel. The first sub-sample was used to assay the soil total nitrogen content and the isotopic analysis after the air-dried soil was sieved through a 0.15-mm sieve, the second sub-sample was used to determine the soil water content, and the third sub-sample was stored in a 4 °C refrigerator as reserve sample.

Plant sampling: The maize aboveground parts in the center of each main plot in an area of 10 m^2^ (4 m long and 2.5 m wide) and all of the maize plants and each micro plot were harvested at maturity for the measurement of the grain yield, the aboveground biomass, and the in-season labeled-N uptake. The leaves (including the sheaths around cobs and stems), the stems (including the tassel), the husk leaves (including the stigma), the grain and the cob cores were separately oven-dried at 75 °C to constant weight to calculate the dry matter. The sub-samples of aboveground parts were processed in the same manner as the soil sample for measurements of the nitrogen concentration and isotopic analysis.

The soil water content (mass basis) was measured using the gravimetric method. The total nitrogen content in the grain and other parts of the maize sub-samples was determined by the micro-Kjeldahl method by digesting the sample in H_2_SO_4_-H_2_O_2_ solution [35]. The soil total nitrogen was measured by the Kjeldahl method. The ^15^N abundance of the soil and plant samples was measured using a Deltaplus XP IRMS (model Isoprime100, Environmental Stable Isotope Lab., CAAS, Beijing, China). The nitrogen uptake by plants was estimated by multiplying the grain and straw dry matter weight by the nitrogen concentration [36].

### 4.4. Calculations and Statistical Analyses

^15^N calculations and tracer recovery [37] (Table 3).

A ^15^N budget was calculated for each plot. The ^15^N uptake by the crops was expressed as the percentage of applied ^15^N fertilizer taken up by the aboveground plant parts and reflects the fertilizer N use efficiency of the plants. The ^15^N retention in the soil was described as the percentage of applied ^15^N fertilizer recovered in the top 60 cm of the soil profile, because the ^15^N abundance reached natural abundance in the 60–70 cm soil layer at harvest. Although nitrogen deposition is a portion of the nitrogen income in croplands, it is difficult to monitor quantitatively due to the limitations of the research conditions. Thus, we did not consider the nitrogen deposition in our study.

The data were subjected to analysis of variance (ANOVA) performed by the SAS 9.2 software with PROC GLM. The means were separated by the least significant differences (LSD) at a probability of 0.05% (the treatment effects were considered for each year), and the figures were drawn with the Sigmaplot 12.5 software.

## 5. Conclusions

Plastic mulching simultaneously increased the maize yield, the grain nitrogen concentration, and the total nitrogen accumulation in the aboveground biomass, and NU_p_E, NPE, and fertilizer nitrogen recovery, compared to the CK treatment. When plastic film was used, the N that accounted for the increase in N uptake by maize was derived not only from the applied fertilizer but also from the soil N, compared to the non-mulching treatment. From the perspective of the fate of fertilizer N, plastic mulching increased fertilizer N retention in the 0–60 cm soil layer and decreased the unaccountable fertilizer N, which reduced the potential risk of nitrogen loss compared with CK.

Our results indicate that plastic film has a positive effect on the fate of labeled-N in maize produced in the rain-fed croplands of the Loess Plateau, but it may not be advantageous for sustainable soil fertility due to the imbalance between the N input and N uptake, which reduces sustainable productivity in rain-fed croplands. Thus, further long-term field experiments are necessary to reveal the effects of long-term plastic mulching on the soil nitrogen balance and forms of nitrogen in the soil, which could provide a scientific basis for the efficient use of nitrogen and the sustainable development of rain-fed agriculture. At the same time, we should use more avenues of N tracking to study nitrogen leaching and gas loss. We can precisely manage fertilizer nitrogen.

## Figures and Tables

**Figure 1 plants-11-02343-f001:**
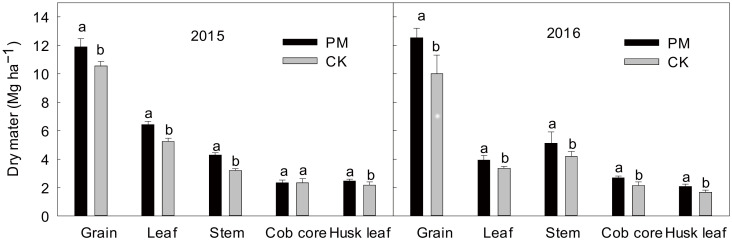
The differences in spring maize dry matter biomass between the PM and M treatments in 2015 and 2016, respectively. The error bars represent the standard error of the mean (*n* = 3). Means followed by different letters are significantly (*p* < 0.05) different between film mulched and non-mulched. PM, maize mulched with plastic film; CK, maize without plastic mulch.

**Figure 2 plants-11-02343-f002:**
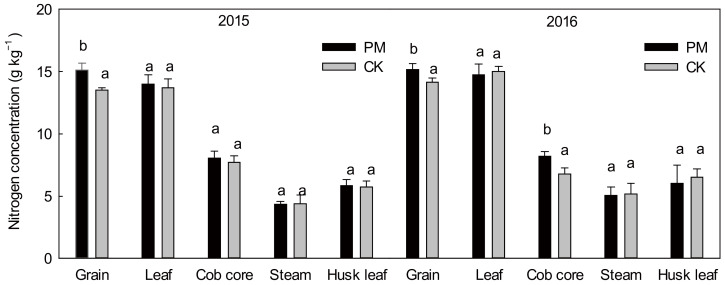
Nitrogen concentration in different aboveground organs under plastic mulching and non-mulching at harvest in 2015 and 2016. Means followed by different letters are significantly (*p* < 0.05) different between film mulched and non-mulched. PM, maize mulched with plastic film; CK, maize without plastic mulch.

**Figure 3 plants-11-02343-f003:**
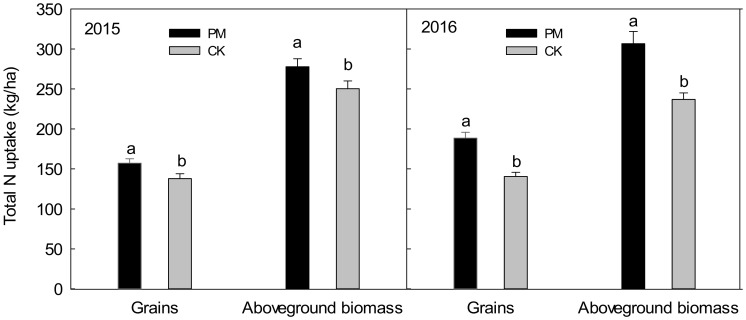
Total nitrogen uptake of the grain and the aboveground biomass under plastic mulching and non-mulching at harvest in 2015 and 2016, respectively. Means followed by different letters are significantly (*p* < 0.05) different between film mulched and non-mulched. PM, maize mulched with plastic film; CK, maize without plastic mulch.

**Figure 4 plants-11-02343-f004:**
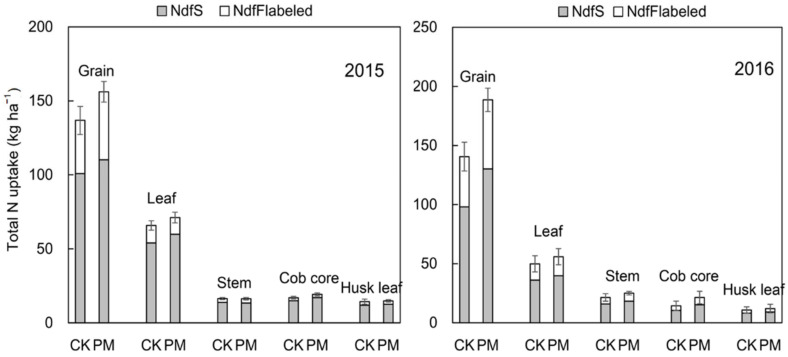
Maize dry matter biomass, total N uptake, N derived from the soil (NdfS) and fertilizer (NdfF_labled_), N derived from labeled-urea fertilizer (NdfF_labeled_), as affected by plastic mulch at harvest in 2015–2016.

**Figure 5 plants-11-02343-f005:**
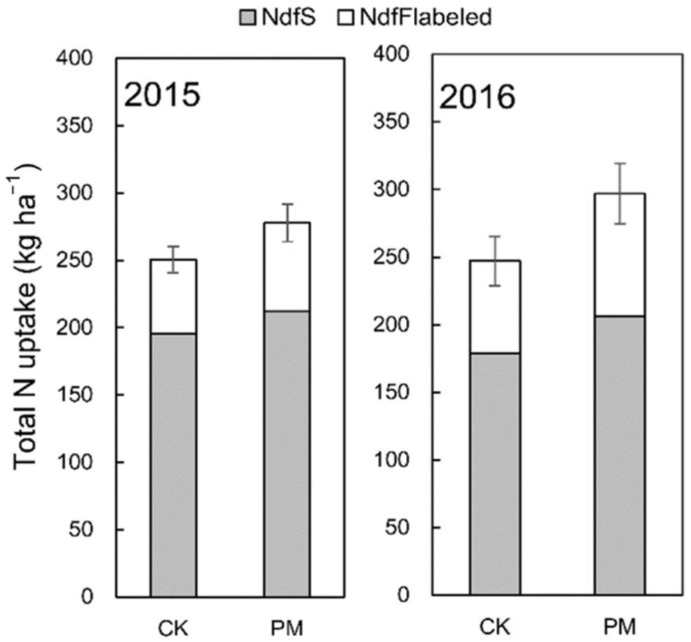
Total N uptake, N derived from the soil (NdfS), and N derived from labeled-urea fertilizer (NdfF_labled_) of aboveground under mulching and non-mulching in 2015 and 2016, respectively.

**Table 1 plants-11-02343-t001:** Nitrogen efficiency under mulching and non-mulching in 2015–2016.

Year	Treatments	NPE	NUpE	NHI	Fertilizer Nitrogen Recovery
		(kg kg^−1^)	(kg kg^−1^)	(%)	(%)
2015	CK	46.80 b	1.11 a	54.65 b	24.29 b
	PM	52.80 a	1.23 a	56.24 b	29.13 a
2016	CK	44.44 b	1.05 b	59.36 a	30.38 b
	PM	55.69 a	1.36 a	61.49 a	40.16 a

Note: Means followed by different letters are significantly (*p* < 0.05) different between film mulched and non-mulched. PM, maize mulched with plastic film; CK, maize without plastic mulch.

**Table 2 plants-11-02343-t002:** Nitrogen budget based on the fate of ^15^N (%) in the top 60 cm of soil at the maturity of spring maize under a fertilizer N rate of 225 kg N ha^−1^ in 2015–2016.

Year	Treatments	Labeled-N in the Aboveground Plant (%)	Labeled-N in the Soil (%)	Unaccounted for Labeled-N (%)
2015	CK	24.3	37.2	37.6
	PM	29.1	48.2	22.6
2016	CK	30.4	35.6	33.9
	PM	40.2	40.9	18.9

**Table 3 plants-11-02343-t003:** ^15^N calculations and tracer recovery.

Abbreviations	Full Name	Calculation
Ndff	N derived from fertilizer N (%)	at% ^15^N excess in plant or soil/at% ^15^N excess in fertilizer × 100
AOTNU	Aboveground organs total N uptake (kg ha^−1^)	aboveground organs dry matter (kg ha^−1^) × N concentration (g kg^−1^)/1000
PNF	Plant N from fertilizer (kg ha^−1^)	AOTNU × Ndff_plant_
RFNS	Residual fertilizer N in the soil (kg ha^−1^)	soil area (ha) × N concentration (g kg^−1^) × soil bulk density (g cm^−3^) × soil thickness (cm) × Ndff_soil_
FNR	Fertilizer nitrogen recovery (%)	PNF/N application rates (kg ha^−1^) ×100
NPE	Nitrogen production efficiency (kg kg^−1^)	Yield (kg ha^−1^)/N application rates (kg ha^−1^)
LNRS	Labeled-N retention in the soil (%)	RFNS/N application rates (kg ha^−1^) ×100
NU_p_E	Nitrogen uptake efficiency	AOTNU/N application rates (kg ha^−1^)
NHI	Nitrogen harvest index (%)	grain dry matter × nitrogen concentration/AOTNU × 100

## Data Availability

The data presented in this study are available on request from the corresponding author.

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
