# Peer review of "Response of the Fate of In-Season Fertilizer Nitrogen to Plastic Mulching in Rainfed Maize Croplands of the Loess Plateau"

_plants, 2022, doi:10.3390/plants11182343_

Round 1

Reviewer 1 Report

Your Materials and methods come after your results and discussion(must fix)

L35 different wording for high

L45 use crop instead of product

L49use better wording for is important (crucial, extensively required, needed etc)

L53seldom? Different wording or make it clear that it has not been studied in regard to your location, region climate soil conditions crop type etc, clarify

L86 what form of nitrogen

L112 2015, 2016 respectively (use this format throughout) helps flow

L136 137 under where? A bit hard to read, flow…

146-148 %,%, an total respectively etc

148 with findings, consistent with finding in current literature, etc…

L169 boosted? Use better word

169-175 cannot make assumptions on enzymes or mineralization in the soil unless you measured it.

L200-207 reword for flow

L227 the word some?  How much in the literature relates to the type of fertilizer leach or volatility in your study?  If the literature is out of context directing relating to your conditions, change or find a better citation.  Just a suggestion.

L305depends on rainfall ad soil water? Precipitation and soil water holding capacity and MPA

L308 for a total plant density/Ha  of 66,667

L308 farmlands, under local current commercial crop production conditions

L339 4.4 Calc……………put in a table or define all the parameters 1st as acronyms as a sub note defining the calculation to make it easier to read like a stats formula

L352 Deposition is a part of a budget.  Need to clarify what quantitatively whats left from you budget that is unaccounted for.

What model statistically?  GLM Proc Mix, etc, regression

Conclusion: there should of used more avenues of N tracking, leach/gas

Author Response

Dear Reviewer,

Thank you for your  comments concerning our manuscript entitled "Response of the fate of in-season fertilizer nitrogen to plastic mulching in rainfed maize croplands of the Loess Plateau" (plants-1769517). We have read your comments very carefully and made the following corrections:

Comment 1: Your Materials and methods come after your results and discussion(must fix)

Response: According to your comments, we adjusted the order of this part.

Comment 2: L35 different wording for high

Response: We revised this sentence.

Comment 3: L45 use crop instead of product

Response: We replaced “product” with “crop”.

Comment 4: L49 use better wording for is important (crucial, extensively required, needed etc)

Response: Thank you for your valuable suggestion. We use “crucial” instead of “important”

Comment 5: L53seldom? Different wording or make it clear that it has not been studied in regard to your location, region climate soil conditions crop type etc, clarify

Response: As your suggestion, we revised this sentence.

Comment 6: L86 what form of nitrogen

Response: The form of nitrogen is total nitrogen.

Comment 7: L112 2015, 2016 respectively (use this format throughout) helps flow

Response: According to your suggestion, we have made revisions throughout the manuscript.

Comment 8:L136 137 under where? A bit hard to read, flow…

Response: We revised this sentence.

Comment 9:146-148 %,%, an total respectively etc

Response: We revised this sentence.

Comment 10:148 with findings, consistent with finding in current literature, etc…

Response: Thank you for your suggestion, we revised this sentence.

Comment 11:L169 boosted? Use better word

Response: We replaced “boosted” with “increased”.

Comment 12:169-175 cannot make assumptions on enzymes or mineralization in the soil unless you measured it.

Response: Thank you for your suggestion. Our team measured soil enzyme. So, we added this reference.

Comment 13:L200-207 reword for flow

Response: According to your comment, we revised this section.

Comment 14:L227 the word some?  How much in the literature relates to the type of fertilizer leach or volatility in your study?  If the literature is out of context directing relating to your conditions, change or find a better citation.  Just a suggestion.

Response: We deleted this sentence.

Comment 15:L305depends on rainfall ad soil water? Precipitation and soil water holding capacity and MPA

Response:Thank you for your suggestion. We revised this sentence.

Comment 16:L308 for a total plant density/Ha  of 66,667

Response: We revised this sentence.

Comment 17:L308 farmlands, under local current commercial crop production conditions

Response: Thank you for your suggestion. We changed this sentence.

Comment 18:L339 4.4 Calc……………put in a table or define all the parameters 1st as acronyms as a sub note defining the calculation to make it easier to read like a stats formula

Response: we revised it.

Comment 19:L352 Deposition is a part of a budget.  Need to clarify what quantitatively whats left from you budget that is unaccounted for.

 Response: Thanks for your suggestion. We will add research and quantification of nitrogen deposition in future research.

Comment 20:What model statistically?  GLM Proc Mix, etc, regression

Response: we revised it.

Comment 21:Conclusion: there should of used more avenues of N tracking, leach/gas

Response: Thank you for your suggestion. We revised this section.

Reviewer 2 Report

The manuscript entitled “Response of the fate of in-season fertilizer nitrogen to plastic mulching in rainfed maize croplands of the Loess Plateau” evaluate the effects of PM on maize yield and N utilization. Author of the manuscript analyzes the effects of plastic mulching on maize crop yield, N content in the grain, and mechanism of N uptake and utilization in maize plants with plastic mulch (PM) and without plastic mulch (CK) on the Loess Plateau detailly. It is a good and meticulous job. However, there are some questions need to be resolve.

1.     The experimental design is too simple. It is lack of control treatment with no nitrogen fertilizer.

2.     In the part of abstract, the author mentioned that “PM increased in-23 season fertilizer N retention in the soil by 11.9-24.8 kg ha-1, and the uncountable fertilizer N de-24 creased by approximately 33.8 kg ha-1 on average”, but we cannot find the soil data in the result part.

3.     It is not clear to reader knowledge can be obtained from this article. Please provide detailed data in the part of “fate of fertilizer N”. The fertilizer nitrogen budget based on the fate of 15N-urea in the upper 60 cm of the soil and labeled-N at maturity. The at% 15N excess in plant or soil should list in the manuscript.

4.     The authors did not monitor dynamic of soil moisture and temperature. Because plastic mulching (PM) improved the root system and increased the soil water content and nutrient uptake by increasing the soil water and temperature, facilitated the transport of more assimilated N into the grain.

Author Response

Dear Reviewer,

Thank you for your comments concerning our manuscript entitled "Response of the fate of in-season fertilizer nitrogen to plastic mulching in rainfed maize croplands of the Loess Plateau" (plants-1769517). We have read your comments very carefully and made the following corrections:

Comment 1:The experimental design is too simple. It is lack of control treatment with no nitrogen fertilizer.

 Response: Thank you for your suggestion. The main purpose of this manuscript is to study the effect of plastic film on N uptake and utilization. Stable isotope tracer technique is a better way to solve this problem.

Comment 2: In the part of abstract, the author mentioned that “PM increased in-23 season fertilizer N retention in the soil by 11.9-24.8 kg ha-1, and the uncountable fertilizer N de-24 creased by approximately 33.8 kg ha-1 on average”, but we cannot find the soil data in the result part.

Response: The soil data really did not appear in the paper. We can deduce these data from Table 3. We added these information in the result part.

Comment 3: It is not clear to reader knowledge can be obtained from this article. Please provide detailed data in the part of “fate of fertilizer N”. The fertilizer nitrogen budget based on the fate of 15N-urea in the upper 60 cm of the soil and labeled-N at maturity. The at% 15N excess in plant or soil should list in the manuscript.

Response: According to your suggestion, we revised this section.

Comment 4: The authors did not monitor dynamic of soil moisture and temperature. Because plastic mulching (PM) improved the root system and increased the soil water content and nutrient uptake by increasing the soil water and temperature, facilitated the transport of more assimilated N into the grain.

Response: Thank you for your valuable suggestion. We did not monitor soil moisture and temperature in this study. In future studies, we will monitor soil moisture and temperature according to your suggestions.